# Structural validation and assessment of AlphaFold2 predictions for centrosomal and centriolar proteins and their complexes

Mark van Breugel [1,2 ✉], Ivan Rosa e Silva [1,2,3] & Antonina Andreeva [2]

Obtaining the high-resolution structures of proteins and their complexes is a crucial aspect of understanding the mechanisms of life. Experimental structure determination methods are time-consuming, expensive and cannot keep pace with the growing number of protein sequences available through genomic DNA sequencing. Thus, the ability to accurately predict the structure of proteins from their sequence is a holy grail of structural and computational biology that would remove a bottleneck in our efforts to understand as well as rationally engineer living systems. Recent advances in protein structure prediction, in particular the breakthrough with the AI-based tool AlphaFold2 (AF2), hold promise for achieving this goal, but the practical utility of AF2 remains to be explored. Focusing on proteins with essential roles in centrosome and centriole biogenesis, we demonstrate the quality and usability of the AF2 prediction models and we show that they can provide important insights into the modular organization of two key players in this process, CEP192 and CEP44. Furthermore, we used the AF2 algorithm to elucidate and then experimentally validate previously unknown prime features in the structure of TTBK2 bound to CEP164, as well as the Chibby1-FAM92A complex for which no structural information was available to date. These findings have important implications in understanding the regulation and function of these complexes. Finally, we also discuss some practical limitations of AF2 and anticipate the implications for future research approaches in the centriole/centrosome field.

[1] Queen Mary University of London, School of Biological and Behavioural Sciences, 4 Newark Street, London E1 2AT, UK. [2] Medical Research Council—Laboratory of Molecular Biology, Francis Crick Avenue, Cambridge CB2 0QH, UK. [3] Present address: University of Campinas, Faculty of Pharmaceutical Sciences, Cândido Portinari Street, Campinas 13083-871, Brazil. ✉email: m.vanbreugel@qmul.ac.uk

An essential aspect in understanding the mechanisms of biological processes and functions is the determination of the three-dimensional structure of the proteins involved in them. Beyond the mechanistic insights into biological activities, high-resolution structural information on proteins also allows rational, structure-based drug design as well as protein engineering to test hypothesis in vivo and to alter protein properties to develop novel functionalities, e.g. the development of activity sensors, biomolecular machines or the engineering of biochemical pathways[1–5].

Experimental methods used to derive the high-resolution structures of proteins and their complexes are currently a bottleneck in supporting progress along these lines. Structure determination by NMR is time-consuming, expensive and mainly limited to small proteins or peptides[6,7]. X-ray crystallography relies on protein crystals that are often challenging to obtain and requires large amounts of purified protein as well as extensive construct screening[8,9]. While offering unique advantages over both of these approaches, cryo-EM is an expensive method and, despite the massive progress in recent years, obtaining high-resolution maps in which atomic models can be build de-novo with high confidence is a time-consuming endeavour[10,11].

As the key information about the protein's three-dimensional shape is deeply encoded into its sequence[12], it should be possible to computationally predict the protein structure from its sequence. Over the past decades, the protein structure prediction field has moved towards achieving this goal, at least for small proteins or protein domains[13]. Building and extending on many years of research provided by this field and utilising powerful and transformative deep-learning approaches, AlphaFold2 (AF2) developed by DeepMind has recently marked a breakthrough in the accuracy of predicting protein structures that has been mirrored by equally impressive achievements by academic groups[14–18]. Besides making the AF2 software freely available to the broad scientific community[17], DeepMind, in collaboration with the European Bioinformatics Institute, made accessible the predicted structural models of the human proteome as well as that of several other organisms[19]. These large-scale predictions along with the availability of the prediction method provide unprecedented opportunity to investigate and explore different aspects of protein structure and organization in a context of full-length proteins. It can be anticipated that they will greatly assist different areas of experimental research and strongly contribute to scientific discoveries.

Focusing on the field of centrosomes, centrioles and cilia, we provide experimental data from our recent work to validate the AF2 structural models and to demonstrate their quality, usability and limitations. We also used the AF2 algorithm to predict and then experimentally verify the TTBK2-CEP164 and Chibby1-FAM92A protein-protein interactions which led to further insights into the functional mechanism and regulation of these complexes. Finally, we discuss some aspects of the AF2 predictions that need improvement and conclude with the impact of AF2 on the centriole/centrosome as well as other fields in life sciences.

## Results and Discussion

**Insights into the domain structure and modular organisation of centrosomal proteins using AF2 prediction models.** CEP192 and CEP44 are proteins with a key role in centrosome and centriole biogenesis. CEP44 was recently identified as a luminal centriolar protein that binds to the A-microtubules (MTs) of centrioles, as well as to the inner centriole protein POC1B[20]. In contrast, CEP192 is an essential component of the pericentriolar material (PCM), a proteinaceous matrix that surrounds centrioles

in which CEP192 exerts a scaffolding and regulatory function[21]. We have recently determined the structures of the N-terminal Calponin Homology (CH) domain of human CEP44 and the so-called Spd2 domain of human CEP192. Both structures were obtained through X-ray crystallography at a resolution of 2.3 and 2.1 Å, respectively, and were experimentally phased (Table 1, Fig. 1, Supplementary Fig. 1). They were not deposited in the Protein Data Bank (PDB) before or at the time when the AF2 protein structure database was made publicly available[19]. The N-terminal domain of CEP44 shares a weak sequence similarity to various CH domains, most of which members of the IFT81/NDC80/Hec families, whereas the Spd2 domain retrieves sequence matches to members of the PapD-like superfamily. In both cases, these weak similarities to protein domains were detectable with HHpred searches[22]. As of the time of writing, the highest-scoring hits for CEP44 CH and CEP192 Spd2 domain were domains from Hec1 (PDB 2IGP, probability: 92.31%, e-value: 1.4, identity: 22%) and from an uncharacterised protein from *P. gingivalis* (PDB 2QSV, probability: 99.53%, e-value: $1.6e^{-11}$, identity: 14%), respectively.

The crystal structure of the CEP44 CH domain showed a typical CH fold consisting of a central four helical bundle elaborated with three short helices. The comparison of the experimental with the AF2 predicted structure (AF-Q9C0F1-F1-model_v1) revealed 116 residues superposed with rmsd of 0.74 Å. (Fig. 1a, b). In contrast, the CEP44 experimental structure superposed with related CH domains with rmsd ranging from 2.8

**Table 1 Crystallographic data collection and refinement statistics.**

|  | SeMet CEP44[1-140] (peak) | SeMet CEP192[1743-2092] (peak) |
|---|---|---|
| **Data collection** | | |
| Space Group | $P4_12_12$ | $P6_1$ |
| Cell dimensions | | |
| a,b,c (Å) | 33.9, 33.9, 243.4 | 104.6, 104.6, 90.2 |
| α, β, γ (°) | 90.0, 90.0, 90.0 | 90.0, 90.0, 120.0 |
| Wavelength (Å) | 0.97928 | 0.97835 |
| Resolution (Å) | 81.12–2.30 (2.38–2.30) [a] | 90.58–2.08 (2.14–2.08) [a] |
| $R_{merge}$ | 0.084 (2.354) | 0.101 (1.311) |
| $R_{pim}$ | 0.019 (0.506) | 0.024 (0.357) |
| I/σI | 17.7 (1.9) | 19.7 (2.5) |
| Completeness (%) | 100.0 (100.0) | 100.0 (100.0) |
| Redundancy | 21.4 (22.0) | 18.6 (14.4) |
| Wilson B-factor (Å²) | 71.6 | 36.8 |
| **Refinement** | | |
| Resolution (Å) | 33.62–2.30 (2.48–2.30) | 90.58–2.08 (2.14–2.08) |
| No. reflections | 7072 (1190) | 32002 (2368) |
| $R_{work}$ / $R_{free}$ | 0.2262/0.2645 | 0.2120/0.2236 |
| Number of atoms | 980 | 2735 |
| Protein | 976 | 2563 |
| Ligand / ion | N/A | 12 |
| Waters | 4 | 162 |
| B-factors | 103.1 | 45.9 |
| Protein | 103.2 | 45.8 |
| Ligand / ion | N/A | 55.1 |
| Waters | 77.5 | 46.6 |
| R.m.s. deviations | | |
| Bond length (Å) | 0.004 | 0.002 |
| Bond Angles (°) | 0.558 | 1.171 |

[a] A single crystal was used for the structure determination. Values in parentheses are for the highest-resolution shell.

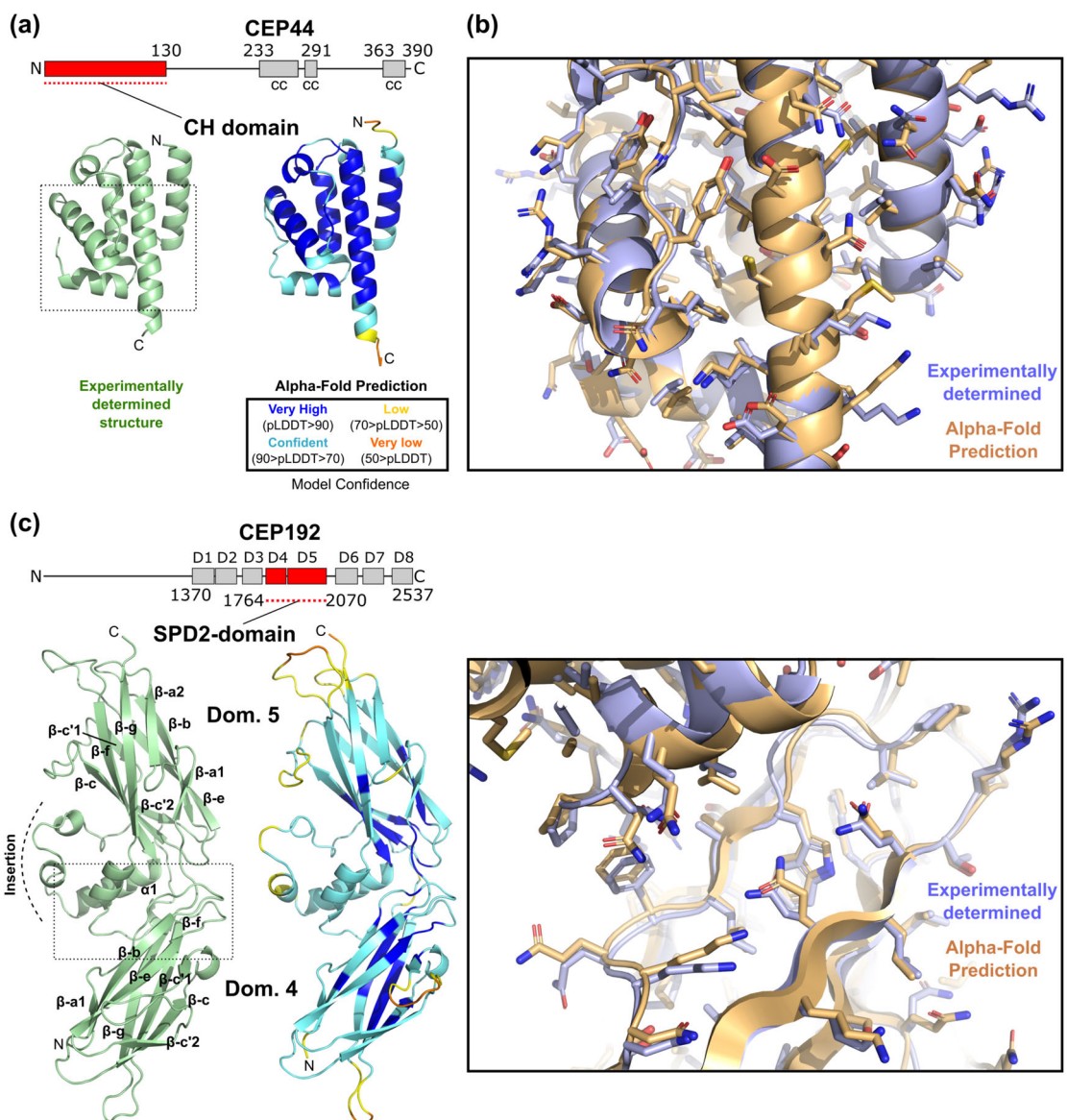

**Fig. 1 AF2 predicts the structures of globular domains of centriolar proteins with high accuracy. a** Ribbon representation of the experimentally determined high-resolution structure of the N-terminal CH domain of CEP44 (PDB 7PT5) and the structure of CEP44's CH domain as predicted by AF2. The AF2 prediction is colour-coded by the pLDDT values indicating the confidence level of the prediction. The dotted box indicates the region shown magnified in panel b. In the domain organisation scheme of human CEP44, cc denotes a predicted coiled coil and amino acid residue numbers indicate domain boundaries. **b** Detailed view of the region boxed in panel a, with the experimentally determined structure (blue) superposed with the AF2 structure prediction (orange). Sidechains are shown as sticks. **c** Comparison of the experimentally determined (PDB 7PTB) and the AF2-predicted structure of the so-called Spd2 domain of human CEP192, presented similarly as in panels a-b. D in the domain overview scheme indicates PapD-like domains. The beta strands of the two subdomains of the Spd2 domain are labelled according to the strand nomenclature of Bullock and colleagues[61].

to 3.1 Å. Thus, the AF2-predicted model is more similar to the experimental structure than any of the homologous templates available in the PDB (Supplementary Fig. 2a). Most of the residues in the predicted structured regions in the AF2 model have very high confidence scores (pLDDT >90, Fig. 1a) that correlate with the model accuracy. The predicted AF2 model has very similar or nearly identical backbone and side-chain conformations and is of comparable quality to the experimental crystal structure (Fig. 1a–b). The CEP44 CH-domain structure revealed the presence of a highly conserved basic patch on its surface (Supplementary Fig. 2b). Mutation of the residues that constitute this patch abolish the CEP44 microtubule and centriole association, rationalising their involvement in microtubule binding[20]. Even in the absence of the experimental CH-domain

structure, the AF2 prediction model of this quality could be successfully used in cryo-EM-based microtubule docking studies or for further mutagenesis design aimed at the identification of additional CEP44 interaction partners. Besides the accurate prediction of the CEP44 N-terminal domain, the AF2 prediction of the alternating coiled coil and unstructured regions that follow the CH domain is in good agreement with other prediction methods and provides an easily accessible model of CEP44's overall domain organisation (Supplementary Fig. 2c).

While insights into the biological function of CEP44's CH domain have been described, the exact functional mechanisms of the CEP192 Spd2 domain are largely unknown. The common isoform of human CEP192 is 2537 residues long and a sequence analysis of its C-terminal part suggests the presence of eight

tandem domains that are similar to members of the PapD-like superfamily. The human CEP192 domains 4 and 5 constitute the so-called Spd2 domain that is ubiquitously present in all SPD2/CEP192 homologues and represents the most conserved region amongst them. Its highest sequence conservation is found in two motifs, as judged by multiple sequence alignment of orthologous sequences: Motif 1 (PLXGYGG) located at the C-terminal end of domain 4, and Motif 2 (GDEXXR) located within domain 5 (Supplementary Fig. 3a).

The crystal structure of the human Spd2-domain (Table 1, Fig. 1c) revealed two seven-stranded Ig-like β-sandwiches with distinctive kinks (β-a1/ β-a2 and β-c'1/ β-c'2) at the edges, a feature that is characteristic for the PapD-like superfamily. The two constituent domains pack together to form an extended cradle-like structure. The two highly conserved motifs PLXGYGG and GDEXXR define the interface between both domains via an extensive network of interactions (Supplementary Fig. 3b). Intriguingly, Motif 2 is found within a large (60 residue) insertion that precedes the C-terminal β-strand g of domain 5 and consists of a helix (α1) and an irregular part that wraps around it before proceeding to complete domain 5 (Fig. 1c). This insertion defines a unique feature that is essential for maintaining the cradle-like conformation of the Spd2 domain.

The comparison of the Spd2 crystal structure with the AF2 predicted model (AF-Q8TEP8-F1-model_v1) revealed striking similarity with 273 structurally equivalent residues superposed with rmsd of 1.83 Å. Despite the fact that most residues have been assigned moderate confidence scores (90 > pLDDT > 70), the AF2 model is accurate with the structures of each individual domain and their relative orientation indistinguishable from the experimentally determined structure (Fig. 1c). Remarkably, the conformation of the unique insertion in the Spd2 domain is also correctly predicted, a feature that the conventional prediction methods fail to predict. It is important to note that the two-domain structure of the *P. gingivalis* homologue (PDB 2QSV) has a very different interdomain interface and relative domain orientation compared to the Spd2 domain (Supplementary Fig. 3c). The individual domains of this structure superposed with the Spd2 domain 4 and 5 with rmsd of 2.48 and 1.88 Å, respectively. None of the known structures of close homologues of the Spd2 domain contain an insertion of comparable size to the Spd2 domain 5 insertion (Supplementary Fig. 3d, e) and therefore the AF2 prediction of the latter is per se a *de novo* prediction.

The AF2 model of full-length human CEP192 also provides the first glimpse into its modular organisation. The C-terminal part of the protein can be described as beads-on-a-string in which eight PapD-like domains are segregated by linkers of different size that probably display a degree of flexibility.

Similarly to the CEP44 CH domain prediction, the outstanding quality of the AF2 model of the Spd2 domain would allow to experimentally address its role in centrosome biology even in the absence of the crystal structure. The two AF2 models of the relatively simple CH domain and the more complex Spd2 domain are both highly accurate and of comparable quality to our experimentally determined structures. As highlighted above, the availability of accurate prediction models of these domains would permit the rationalisation of cell biological and biochemical findings, the generation of structure-based hypotheses and mechanistic studies based on them.

**Insights into the molecular details of known centrosomal protein complexes**. CEP164 is a key component of basal bodies and its interaction with TTBK2 is essential for ciliogenesis and thereby for human life. We have recently demonstrated how the N-terminal domain (NTD) of CEP164, consisting of a WW domain and a three-helical bundle, recruits TTBK2 to the centriolar distal appendages through engaging a C-terminal Pro-rich sequence (TTBK2$^{1074-1087}$)[23]. This recruitment allows the phosphorylation of several proteins at centriole distal ends by TTBK2 which sets in motion the downstream events of the complex cilia formation program[23–28].

While the high-resolution structure of this complex is known[23] (PDB 7O3B), several questions concerning this complex remain unanswered. It is unclear what determines the TTBK2 binding specificity given the abundance of Pro-rich sequences in other proteins. Furthermore, it is also unknown whether the alpha-helical bundle that stabilises the CEP164 WW domain, also makes a direct contribution to complex formation. The TTBK2 sequence contains a highly conserved segment downstream of the Pro-rich region (TTBK2$^{1088-1098}$, Fig. 2a, Supplementary Fig. 4a) that could potentially participate in CEP164 binding. To test this hypothesis, we exploited *in silico* an approach frequently used in crystallography. We fused through a flexible linker the sequence of the corresponding fragment of TTBK2 to the CEP164-NTD and subjected the chimeric sequence to AF2 prediction.

The comparison of the predicted CEP164-TTBK2 "fused" complex with the experimental structure showed a remarkable similarity of its known features, such as the CEP164-NTD fold, as well as its contacts with the Pro-rich region of TTBK2 (Fig. 2b). The predicted structure revealed that the sequence downstream of the TTBK2 Pro-rich region makes extensive interactions with the conserved interface between the CEP164 WW and helical domains (Fig. 2b, Supplementary Fig. 4a–c). These interactions are mainly electrostatic with Arg1085 and Arg1091 engaging in salt bridges with Asp85 and Asp89, respectively. In addition, Tyr1095 forms a hydrogen bond with His86. Several residues make hydrophobic contacts including Tyr1095 and Leu1092 from TTBK2 and Pro64 and Ile72 from CEP164. Interestingly, the experimental structures of CEP164-NTD were obtained via co-crystallization with two different nanobodies that both engaged with CEP164 Asp89 through an arginine in their CDR3 loop[23]. Thus, the predicted salt bridge between TTBK2 Arg1091 and CEP164 Asp89 appears to mimic the nanobody interaction. During the review of this manuscript, AlphaFold-Multimer was released[29] that extends AlphaFold2 to multiple chain predictions. The predicted CEP164-TTBK2 complex using AlphaFold-Multimer was essentially the same as the predicted "fused" complex" with very small differences in the conformation of some side-chains. One of these was the side-chain of Arg1076 that in the AlphaFold-Multimer model does not engage in any interactions while in the predicted CEP164-TTBK2 "fused" complex it forms a hydrogen bond with the Gly80 O. In the experimental structure, this residue is involved in a salt bridge with Asp10. We have previously shown the importance of Arg1076 for CEP164-TTBK2 complex formation[23]. While in the predicted CEP164-TTBK2 "fused" complex, the Arg1076 alternative conformation can still support its role in stabilizing the complex, the conformation of this residue in the AlphaFold-Multimer model does not provide any clues to its possible function.

We tested the importance of the extended CEP164 binding region of TTBK2 using mutagenesis. We targeted three distinct regions for mutagenesis (TTBK2$^{1088-1090}$, TTBK2$^{1091-1094}$ and TTBK2$^{1095-1098}$) in which we substituted consecutive residues in TTBK2 to alanine. Subsequently, we performed pull-down experiments with 3xFLAG tagged CEP164 and the 3×HA tagged TTBK2 constructs, that were transiently overexpressed in human tissue culture cells. The results shown in Fig. 2c suggest that the predicted interactions indeed contribute to the binding affinity, with the mutation of TTBK2$^{1091-1094}$ having a pronounced effect on complex formation (Supplementary Fig. 5a). These findings

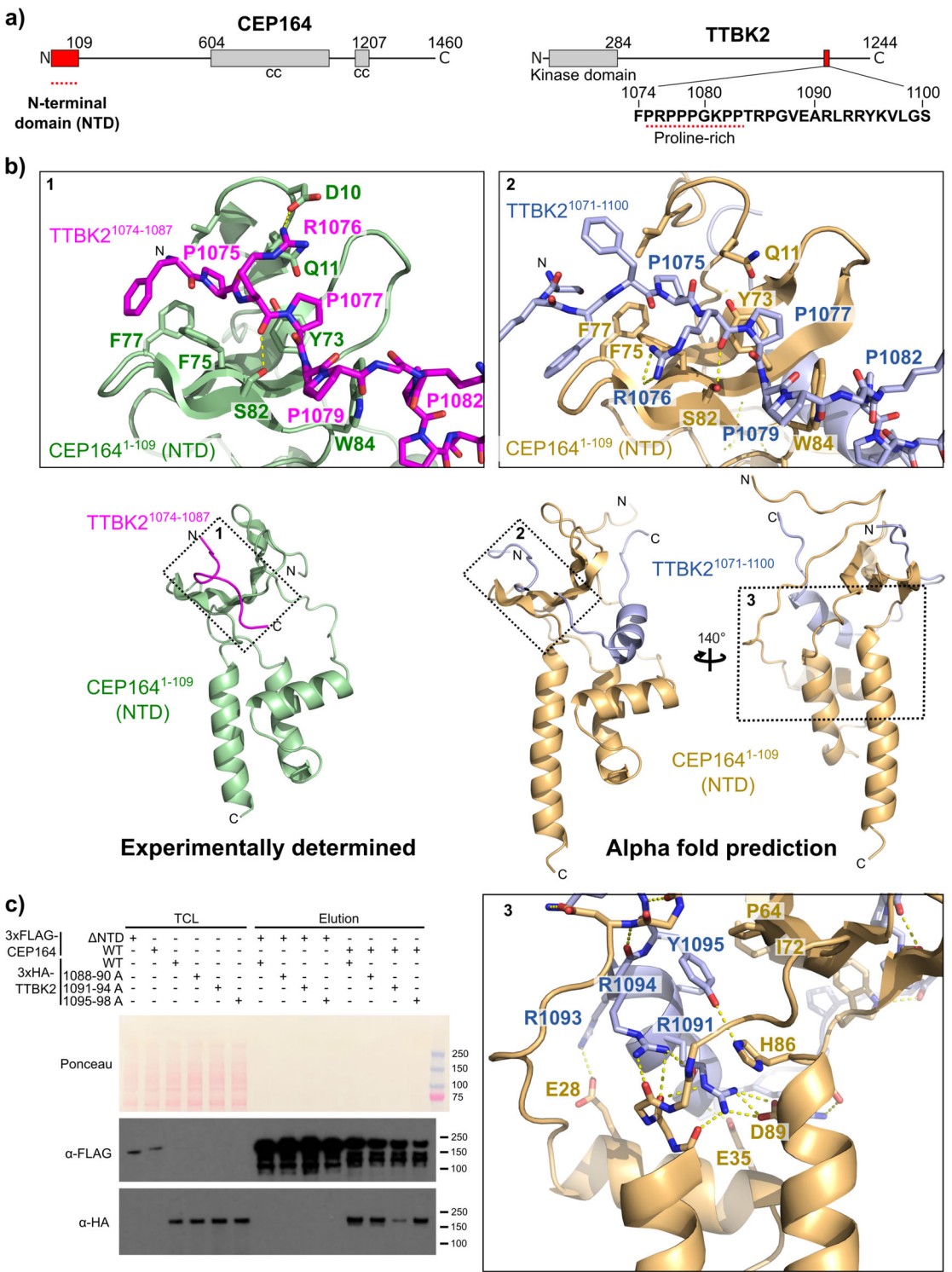

**Fig. 2 AF2 predictions identify an additional interface in the CEP164-TTBK2 complex. a** Domain organisation of human CEP164 and TTBK2. cc, coiled coil. Amino acid residue numbers indicate domain boundaries. The amino acid sequence of the CEP164 N-terminal domain (NTD)-binding region of TTBK2 is shown under the TTBK2 scheme. The proline-rich sequence that engages the WW domain of CEP164 NTD is highlighted by a red-dotted line. **b** Ribbon representation of the experimentally determined high-resolution structure[23] (PDB 7O3B) of the CEP164[1–109]-TTBK2[1074–1087] complex and the AF2-predicted structure of the CEP164[1–109]-TTBK2[1071–1100] complex. Where applicable, the rotation angles of the views are indicated. Dotted numbered boxes highlight the regions that are shown magnified above and below the overview panels. In these magnifications, selected sidechains are shown as sticks and are labelled. Dotted yellow lines designate hydrogen bonds. **c** The AF2-predicted additional TTBK2-CEP164 interface is important for efficient complex formation. Western blot showing a pull-down experiment with lysates from cells expressing the indicated 3xFLAG-tagged CEP164 or 3xHA-tagged TTBK2 constructs. TCL: Total cell lysate.

might also explain why the binding affinity of the TTBK2[1074–1087] region alone was found to be lower than that of full-length TTBK2[23]. Consistent with this notion, using the longer fragment TTBK2[1–1099], instead of the TTBK2[1–1087] truncation, restores the CEP164 binding affinity to full-length TTBK2 levels in pulldown experiments (Supplementary Fig. 5b).

Phosphatidylinositol-4-phosphate (PtdIns4P) was proposed to bind to the region containing TTBK2 residues R1091, R1093 and R1094 to down-regulate CEP164 engagement by TTBK2 and, correspondingly, the deletion of this region was found to strengthen the TTBK2-CEP164 interaction[30]. Our biochemical data together with the associated AF2 model provide valuable insights and create hypotheses that can be further explored experimentally to test this proposed regulatory mechanism. The structure prediction suggests that the aforementioned residues interact with CEP164 and, consistent with this model, their mutation leads to a considerably weakened interaction. If the CEP164-TTBK2 complex is "breathing" and the residues R1091, R1093 and R1094 are intermittently available for PtdIns4P binding, the binding competition between PtdIns4P and CEP164 could indeed regulate the binding affinity and complex formation in the biologically critical process of cilia formation. However, further research will be necessary to structurally confirm the AF2 prediction and to establish whether PtdIns4P binding can affect the identified CEP164-TTBK2 interface and regulate protein complex formation and ciliogenesis in vivo.

**Predicting the structure of protein complexes for which no experimental structure is available**. We tested if AF2 can predict the structure of protein complexes that were identified biochemically, but for which no experimental structural information is available to date. We focused on the Chibby1-FAM92A complex that is essential for faithful cilia formation[31]. While Chibby1 anchors this complex to the basal body, possibly through DZIP1L and CEP164[32,33] at the distal appendages, FAM92A, via its BAR domain, likely binds to lipid membranes[31,34]. Lipid membrane binding and remodelling is a crucial process in ciliogenesis[35], explaining the biological importance of this complex.

The AF2 prediction models of the FAM92A (AF-A1XBS5-F1-model_v1) and Chibby1 (AF-Q9Y3M2-F1-model_v1) full-length proteins indicate that the C-terminal region of FAM92A and the N-terminal region of Chibby1 are probably largely unstructured. In addition to its N-terminal unstructured region, Chibby1 is predicted to contain a small three-stranded antiparallel beta-sheet domain and a C-terminal coiled coil region (Fig. 3a). Prior to the protein complex prediction with AF2, we fused *in silico* the Chibby1 sequence through its N-terminus to the C-terminus of FAM92A. Since both FAM92A and Chibby1 were reported to form homodimers[31,36] we modelled the fusion protein as dimeric. Predicting the 2:2 FAM92A-Chibby1 complex from its non-fused chains using the released AlphaFold-Multimer algorithm[29] resulted in model predictions that were essentially the same. In the predicted AF2 model (Fig. 3a, Supplementary Fig. 6a, b), the N-terminal unstructured region of Chibby1 makes extensive contacts on the concave side of the FAM92A BAR domain. In contrast, the beta-sheet domain of Chibby1 docks onto the convex side of the BAR domain through a conserved hydrophobic cage assembled from Pro43, Phe54 and Trp59 that accommodate Met192 of FAM92A (Fig. 3a, Supplementary Fig. 6c). Additional hydrophobic contacts are made by Leu199 and Met45 from FAM92A and Chibby1, respectively. The top five ranked predictions differ in the relative orientation of the C-terminal coiled-coil stalk and the beta-sheet domain of Chibby1 (Fig. 3a), suggesting that the linker between them is flexible allowing the

FAM92A BAR domain to explore different orientations and optimise its contacts with membranes.

To experimentally verify the AF2 predictions and establish the contributions of these putative interfaces, we mutated the FAM92A-associated Chibby1[1–22] region (1-MPFFGNTFSPKKTPPRKSASLS-22 to 1-AAAAAAAAGADDAGADDAGAAA-22). In addition, we also replaced the hydrophobic interface residues in the Chibby1 beta-sheet domain (M45, L47 or W59) and in FAM92A (M192 or L199) to glutamate. We transiently overexpressed in tissue culture cells these constructs as 3×FLAG- (Chibby1) or 3xHA-tagged (FAM92A) proteins and performed pull-down experiments with the corresponding cell extracts. In agreement with the predicted structural model (Fig. 3a), the mutation of the Chibby1 and the FAM92 interface residues, severely compromised complex formation (Fig. 3b, c, Supplementary Fig. 5c, d). In contrast, the mutation of the 1-22 region of Chibby1 did not have such a strong impact on binding, suggesting that it is probably more weakly associated. Intriguingly, 14-3-3ζ (YWHAZ) engages the Chibby1[1–22] region in a phosphorylation-dependent way[37] and the Chibby1[Δ1-22] truncation was found to abolish FAM92A binding[31]. However, our mutational analysis suggests that the binding contribution of this region is smaller than the truncation experiments indicated. Further experiments will be required to evaluate whether and how Chibby1 phosphorylation and 14-3-3 binding can impact on FAM92A engagement with Chibby1 and cilia formation in vivo.

Together, the insights from the AF2 predictions delineate the structural basis of the Chibby1-FAM92A interaction, identify the major interface involved in binding and provide important information on key aspects of cilia formation. The predicted model suggests that the membrane-binding surface of FAM92A is facing away from Chibby1 and would therefore be presented at the distal appendages in an orientation that maximises the potential of FAM92A for membrane interaction. Thus, the coiled-coil stalk of Chibby1 can serve as an anchor and at the same time allow the bound FAM92A BAR domain to explore different orientations. This property could be crucial during the membrane remodelling events that occur during cilia formation.

**Limitations of using AF2 protein structure predictions**. The AF2 prediction models vary in their accuracy and quality but come with a reported confidence level, i.e. the pLDDT score. This score is a good guide for the model quality and it correlates well with the accuracy of the models when assessed against our experimentally determined structures. However, as highlighted above, further experimental characterisation and structure-based mutational studies are required to validate the structural models. Particularly, this is essential when the AF2 algorithm is used to predict the structure of protein complexes as the general ability of AF2 to accurately predict protein-protein interactions remains to be determined.

Despite the important insights that can be readily obtained with AF2, there are some limitations that can currently restrict its utility in solving biological problems. For example, the vast majority of centrosome/centriole proteins contain coiled-coil domains. Some of these proteins are even entirely composed of coiled coils and they play important roles as architectural and functional modules. Without a prior knowledge of whether these tend to self-associate or engage in heteromeric assemblies, the AF2 algorithm appears to have difficulty with their prediction. The single chain-based predictions lack structural plausibility as judged by the models of, for example, CEP250, CEP135, CEP290, CEP131 and other centriolar or centrosomal proteins available from the AF2 database. Similarly, our attempts to model heterodimeric or multimeric coiled-coil assemblies resulted in predictions that lacked plausibility and were of limited utility. One possible reason for the AF2 under-

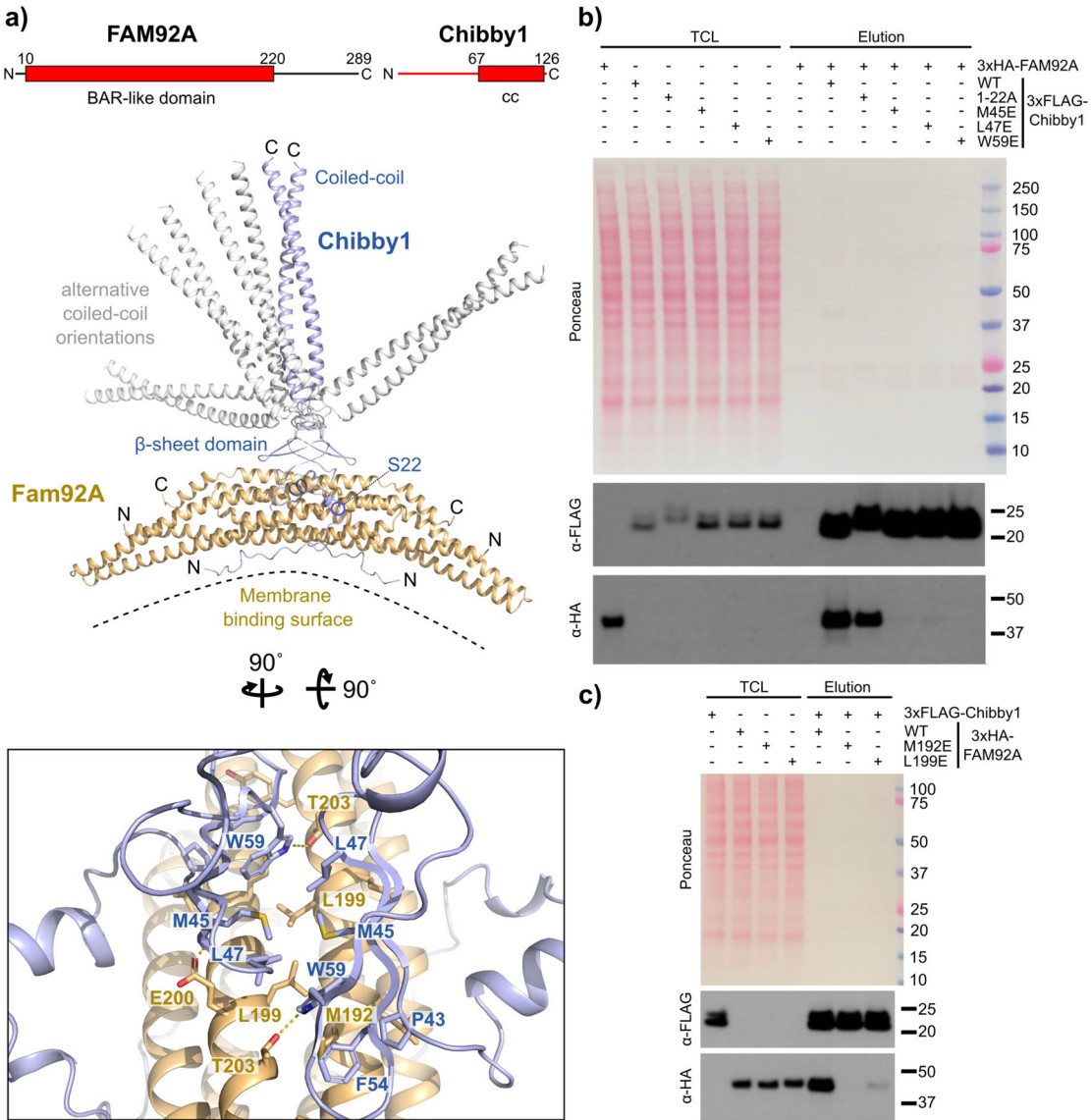

**Fig. 3 AF2 predicts the structure of the FAM92A-Chibby1 complex. a** Ribbon representation of the structure of the human FAM92A-Chibby1 complex as predicted by AF2. The top five predictions were virtually identical in the FAM92A BAR domain and the Chibby1 beta-sheet domain and mainly differed in the orientation of Chibby1's coiled-coil domain. The different coiled-coil orientations are shown for all five models (in blue or grey). Shown boxed is a detailed view on the interface between the beta-sheet domain of Chibby1 and the BAR domain of FAM92A. Selected sidechains are shown as sticks and are labelled. Dotted yellow lines designate hydrogen bonds. The rotation angles to obtain this view are indicated. In the domain organisation scheme of FAM92A and Chibby1, cc denotes a predicted coiled coil and amino acid residue numbers indicate domain boundaries. Coloured in red are the protein parts shown in the structural models. **b** The AF2-predicted interface between the Chibby1 beta-sheet domain and the FAM92A BAR domain is essential for complex formation. Western blot showing a pull-down experiment with lysates from cells expressing 3xHA-tagged FAM92A or the indicated 3xFLAG-tagged Chibby1 constructs. TCL: Total cell lysate. **c** As in panel b, but pulldown with 3xFLAG-tagged Chibby1 and the indicated 3xHA-tagged FAM92A constructs.

performance may be due to under-representation of experimentally determined coiled-coil structures that are available in the PDB. According to the SCOP 2 release 2021-12-17[38], there are 470 representative domains in the fibrous proteins category, containing mainly coiled-coil domains, compared to 34,094 globular and 1119 membrane representative domains. Thus, although SCOP 2 does not entirely cover the PDB, it clearly reveals the disproportionality of the structural data. Another explanation for the AF2 under-performance may be hidden in the known binding promiscuity of these domains and their complex modes of oligomerisations. This reinforces the need to gather more experimental data on these assemblies and, apparently, to further improve the corresponding prediction algorithms.

Another limitation of AF2 is that the predictions yield a static picture of the structure and contain little or no information on its dynamics. Conformational changes in proteins are often a key to their function and regulation and are frequently triggered by binding to other molecules (e.g. proteins or small molecules/ ligands) or by post-translational modifications. The current prediction algorithm is unable to model these and is therefore largely unable to provide a crucial aspect of understanding the protein structure-function relationship. The alternative orientations of the C-terminal coiled-coil stalk relative to the beta-sheet domain of Chibby1 (Fig. 3a) seen in the top-ranking AF2 models, for example, could suggest flexibility of the Chibby1-FAM92A complex. While these differences in the prediction models can

provide a basis for speculation, they are not sufficient to apprehend the conformational dynamics of this complex.

The prediction of the relative orientation of domains in multidomain proteins and their interdomain contacts also appears to be challenging, as pointed out previously[39]. This prediction is particularly problematic for multidomain proteins in which individual domains are segregated by unstructured regions. In addition to the interdomain geometry, the polypeptide chain of the unstructured regions frequently threads and adopts conformations that lack plausibility. For example, the AF2-predicted full-length CEP192 model (AF-Q8TEP8-F1-model_v1) available from the AF2 database has several problematic areas. The polypeptide chain near the c'1-c'2 strand switch of domain 2 threads through the insertion in domain 5 and comes back to complete the β-sandwich of domain 2. This entanglement creates a knot in the polypeptide chain. While knots have been observed in proteins, they are exceedingly rare[40] and are usually highly conserved across related proteins, in contrast to the knot in the predicted human CEP192 full-length model that is probably a result of erroneous prediction. We used AF2 to predict the structure of a fragment of CEP192 that spans six consecutive domains (domain 1-6) and has a length of 893 amino acid residues. While the predicted structure of the Spd2 domain (domain 4-5) was essentially the same, we could not identify any knots, unlike in the full-length CEP192 model available from the AF2 database. Thus, the problem of the polypeptide chain entanglement might be due to either the length of the predicted full-length CEP192 protein (2537 amino acid residues) or the exact way the models in the AF2 database have been computed. Thus, while the prediction of individual domains in the database can be of outstanding quality, there might be problems in predicting the spatial multi-domain arrangement and inter-domain interactions in full-length proteins, as pinpointed by the full-length CEP192 model. These predictions should be always used and interpreted with caution.

## Conclusion

This paper is an attempt to showcase some of the capabilities of AF2 to provide physiologically, structurally and biochemically plausible structures of protein domains and their complexes. Through a number of compelling examples of centriolar and centrosomal proteins, supported by our experimental data, we were able to validate and demonstrate the quality and usability of AF2 structural models. Importantly, we showed that AF2 can not only generate a three-dimensional structure of protein domains to atomic detail but can also correctly predict a novel interdomain interface and de novo build a large insertion region in the structural model. Our work illustrates that AF2 can be broadly applicable to centrosomal research and beyond, and can help to provide insights into biologically important processes and speed up hypothesis-driven research aimed at understanding their mechanisms and regulation. Thus, we anticipate that AF2 will have a notable impact on the centrosome as well as other fields in life sciences and will help to accelerate the process of converting structural knowledge into biological or biotechnological insights.

## Methods

**Pull-downs with 3xFLAG constructs**. Flp-In T-REx 293 cells (gift from Dr. Ramanujan Hegde, MRC-LMB, Cambridge, UK) were transfected individually with pcDNA 3.1 derivative vectors expressing (from a CMV promoter) 3xFLAG-tagged human Chibby1 (Uniprot Q9Y3M2) or CEP164 (Uniprot Q9UPV0, natural variant T988S) or 3xHA-tagged human FAM92A (Uniprot A1XBS5) or TTBK2 (Uniprot Q6IQ55, natural variant L8P) constructs using Polyethylenimine (PEI MAX, MW 40000, Polysciences). Cells were lysed by sonication in lysis buffer (PBS, 0.1% (v/v) IGEPAL CA-630, supplemented with Complete Protease Inhibitor (EDTA free, Roche)) and lysates were cleared by centrifugation. Cleared lysates containing the 3xFLAG-tagged constructs were added to anti-FLAG M2 magnetic beads (M8823,

Sigma). After incubation on a rotator for 1 h at 4 °C, beads were washed with lysis buffer and the cleared lysates containing the 3×HA-tagged constructs were added to the beads. After another incubation on a rotator for 1 h at 4 °C, beads were washed with lysis buffer, eluted with SDS and eluates subjected to Western blotting using a mouse monoclonal anti-FLAG antibody (F1804, Sigma) or a polyclonal rabbit antibody against the HA-tag (kind gift of Dr. Ramanujan Hegde, MRC-LMB, Cambridge, UK). All experiments have been performed in independent duplicates.

**Recombinant protein purification**. DNA encoding human CEP44[1−140] (Uniprot Q9C0F1) or human CEP192[1743−2092] (Uniprot Q8TEP8) were cloned into vector pFASTBac HTa or pACE Bac1 and were N-terminally tagged either with His$_6$ (CEP44) or the His$_6$-lipoyl domain from *Bacillus stearothermophilus* dihydrolipoamide acetyltransferase[41] (CEP192). These tags can be removed by cleavage with TEV protease (CEP44) or PreScission protease (CEP192), leaving the amino acid residues GAMDP or GP on the N-terminus of the corresponding protein, respectively. Baculoviruses were obtained from these constructs using standard procedures and used to infect *Sf*9 insect cell suspension cultures in ESF921Δ, Methionine deficient medium (Oxford Expression Technologies) at a cell density of ~1 × 10$^6$ cells / mL at 27 °C. 10, 24 and 48 h post-infection L-Selenomethionine was added to the culture to a concentration of 0.1 mg/ml and the cells harvested 72 h post-infection. Proteins were purified from cell lysates (prepared by sonication and centrifugational clearing) by Ni-NTA (Qiagen) chromatography. Purified TEV protease (gift from Mark Allen, MRC-LMB, Cambridge, UK) or GST-PreScission protease (CEP192) were added and the eluates dialysed against 50 mM Tris-Cl pH 8.0, 500 mM NaCl, 5 mM imidazole pH 7.6, 10 mM beta-mercaptoethanol. The cleaved eluates were incubated with Ni-NTA (Qiagen) resin and the flow-throughs purified further by gel-filtration on a Sephacryl S-300 column (GE Healthcare) run in 10 mM Tris-Cl pH 8.0, 500 mM NaCl, 3 or 4 mM DTT. Peak fractions were concentrated, snap frozen in small aliquots and stored at −80 °C.

**Protein crystallisation**. CEP44[1−140] crystals were obtained by the vapour diffusion method at 19 °C using 100 nl protein solution and 100 nl of reservoir solution which was 0.1 M Hepes pH 7.3, 10% isopropanol, 10% PEG-4000. Crystals were mounted after six days in 0.1 M Hepes pH 7.5, 7.5% isopropanol, 30% PEG-400, and frozen in liquid nitrogen.

CEP192[1743−2092] crystals were obtained by the vapour diffusion method at 19 °C using 100 nl protein solution and 100 nl of reservoir solution which was 0.1 M Na-Acetate pH 5.4, 3.7 M ammonium nitrate. Crystals were mounted after 1 day in 0.1 M Na-Acetate pH 4.6, 1 M ammonium nitrate, 30% glycerol and frozen in liquid nitrogen.

**X-ray crystallography data processing**. Diffraction datasets were collected at 100 K from the flash-frozen protein crystals using synchrotron radiation at the Diamond Light Source (Didcot, UK) at beamline I03 (CEP44) and I24 (CEP192) to a resolution of 2.3 Å and 2.1 Å, respectively. Further details of the dataset collections and analyses are provided in Table 1. Datasets were integrated using iMOSFLM[42] and were scaled using AIMLESS[43]. The structures of CEP44[1−140] and CEP192[1743−2092] were solved by single-wavelength anomalous diffraction (SAD) using the CRANK2 pipeline[44] (CEP44) or SHELX C/D/E[45] (CEP192) and the models constructed by cycles of refinement in REFMAC[46] or PHENIX.REFINE[47] and manual building in COOT[48]. Refinement statistics of the final models are summarised in Table 1. Model quality statistics, as judged by analysis with MolProbity[49], were for the CEP44 and the CEP192 structure: Overall score 1.09 and 1.09, clashscore 3.04 and 2.67, poor rotamers 0.89% and 0.34%, Ramachandran favoured residues 98.28% and 98.10%, Ramachandran outliers of 0 and 0%, respectively. Structure visualization was done in PyMOL (The PyMOL Molecular Graphics System, Version 2.0 Schrödinger, LLC). Coordinates and structure factors have been deposited at the PDB under code 7PT5 (CEP44) and 7PTB (CEP192).

**AF2 structure prediction and structure comparisons**. The predicted structures of human CEP44[1−130] and CEP192[1763−2065] were obtained from the respective full-length structure predictions as provided in the AlphaFold protein structure database hosted by the EBI (https://alphafold.ebi.ac.uk)[19].

The predicted structures of the human TTBK2[1071−1100]-CEP164[1−109] complex where obtained from the sequence of TTBK2[1055−1100] fused to the N-terminus of CEP164[1−109] with the linker GAGSGAGS separating both. This fusion sequence was submitted to the AlphaFold Colab at https://colab.research.google.com/github/deepmind/alphafold/blob/main/notebooks/AlphaFold.ipynb before the release of AlphaFold-Multimer[29]. Similar results were obtained when the sequence of TTBK2[1001−1244] and the sequence of CEP164[1−109] were submitted as separate chains (using a 1:1 homo-oligomer setting and ranking by pTMscore) to the AlphaFold2_advanced Colab at https://colab.research.google.com/github/sokrypton/ColabFold/blob/main/beta/AlphaFold2_advanced.ipynb and also when the sequence of TTBK2[1054−1244] and the sequence of CEP164[1−109] were submitted as separate chains to AlphaFold-Multimer[29] once it was released at https://colab.research.google.com/github/deepmind/alphafold/blob/main/notebooks/AlphaFold.ipynb.

The predicted structures of the human FAM92A$^{1-228}$-Chibby1 complex where obtained by submitting the sequence of full-length FAM92A, fused to the N-terminus of full-length Chibby1 to the ColabFold: AlphaFold2 w/MMseqs2 Colab at https://colab.research.google.com/github/sokrypton/ColabFold/blob/main/AlphaFold2.ipynb (v1.0)[50] using the default settings and the homooligomer option set to 2 (dimer). Similar results were obtained when the sequence of full-length human FAM92A and the sequence of full-length human Chibby1 were submitted as separate chains (using a 2:2 homo-oligomer setting and ranking by pTMscore) to the AlphaFold2_advanced Colab at https://colab.research.google.com/github/sokrypton/ColabFold/blob/main/beta/AlphaFold2_advanced.ipynb and also when submitted to AlphaFold-Multimer[29] once it was released at https://colab.research.google.com/github/deepmind/alphafold/blob/main/notebooks/AlphaFold.ipynb.

The CEP192 structure comprising its PapD-like domains 1-6 was predicted with the AlphaFold Colab at https://colab.research.google.com/github/deepmind/alphafold/blob/main/notebooks/AlphaFold.ipynb. This Colab does not use templates.

Structure comparisons were performed using TopMatch[51].

The multiple sequence alignment of the Spd2 domain was produced with MAFFT[52], manually corrected and visualized using Jalview[53]. This alignment was used to compute the conservation scores using ConSurf[54] that were then mapped on the structure of the human CEP192 Spd2 domain.

The secondary structure and coiled-coil regions were predicted using PsiPred[55], Jpred[56], and Marcoil[57], Deepcoil[58] and Pcoils[59,60].

**Statistics and Reproducibility**. All pulldown experiments were performed in independent duplicates ($n = 2$). Replication was successful and consistent.

**Reporting summary**. Further information on research design is available in the Nature Research Reporting Summary linked to this article.

## Data availability

The data that support the findings of this study are presented in this manuscript and are available from the corresponding author upon request. The coordinates and structure factors of the CEP44 and CEP192 structure have been deposited at the PDB under code 7PT5 and 7PTB, respectively.

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

## Acknowledgements

We acknowledge Darren Sherrell (I24, MX15916-56) and Neil Paterson (I03, MX15916–59) at the Diamond Light Source (UK) for beamline support and the X-ray crystallography facility from the MRC-LMB. We thank Dr. Ramanujan Hegde (MRC-LMB, Cambridge, UK) for Flp-In T-REx 293 cells, the pcDNA3.1 derivative (3×FLAG / 3xHA) plasmids, and the rabbit anti-hemagglutinin polyclonal antibody. This work was supported by the Medical Research Council (file reference MC_UP_1201/3) and the Queen Mary University of London (both to M.v.B.).

## Author contributions

I.R.S. and M.v.B. determined the crystal structures. M.v.B. performed the biochemical experiments. M.v.B. and A.A. performed the structure predictions, A.A. contributed the bioinformatics and structural analyses. M.v.B. and A.A. wrote the manuscript with contributions from I.R.S. All authors contributed to the reviewing and editing of the manuscript.

## Competing interests

The authors declare no competing interests.
