## [Peer Review File · Communications Biology]

Reviewers' comments:

Reviewer #1 (Remarks to the Author):

The manuscript entitled "Alphafold2 and centrosome research: a tale of strengths and limitations" by Breugel, et al describes the application of using AlphaFold2-predicted protein structures for biochemical studies of centrosome proteins. Using X-ray crystallography, the authors validated the accuracy of AlphaFold-predicted structures of CEP44 CH domain and CEP192 domains 4 and 5. They extended their recent structural work on the complex between CEP164 NTD and TTBK2 proline-rich domain, from which they found previously unknown interactions using AlphaFold. They validated such interactions through site-directed mutagenesis. To show whether AlphaFold can generate hypothesis for studying protein/protein interactions, they predicted a complex structure between FAM92A BAR-like domain and Chibby1 coiled-coil domain and validated their interactions using a pull-down assay. Through three examples of six centrosomal proteins, they provide convincing cases that AlphaFold may be broadly applicable for centrosome research. The manuscript is well written, although there are quite a few typos through the text. The work is technically sound and may be of interest to structural biologists, biochemists, and cell biologists in designing and optimizing their experiments. I have some suggestions that need to be addressed prior to acceptance for publication.

1) Results & Discussion, p 4, lines 2-5: The authors state "Besides the accurate prediction, ... with other prediction methods and provides an easily accessible model of CEP44's overall domain organisation." What are the other prediction methods and how were these models compared? The authors should at least give some numbers to justify such a statement.

2) Results & Discussion, p 4, second paragraph: " Its most significant sequence conservation is found in two motifs, as judged by multiple sequence alignment of orthologous sequences: Motif 1 (PLXGYGG) located at the C-terminal end of domain 4, and motif 2 (GDEXXR) located within domain 5." Missing either a reference or a multiple sequence alignment to show the conservation of the two motifs. Alternatively, the authors can map the conservation to a surface or a ribbon representation, similar to what they did for CEP44 CH domain (Figure S1A) or CEP 164/TTBK2 (Figure S2D).

3) Results & Discussion, p 5, lines 2-9: "While the structure of each individual domain is likely in contrast to the knot in the predicted human CEP192 full-length model that is probably a result of erroneous prediction". Please consider moving this discussion to the section of "Limitations of using AF2 protein structure predictions".

4) Results & Discussion, p 7, second line from bottom: Please define "14-3-3" when first time mentioned.

5) Conclusion & Outlook: In this reviewer's option, the authors should focus on Conclusion only. Their Outlook statements are not related to the work they present, and perhaps should be deleted.

6) Materials & Methods, Protein crystallization: Please correct typos: "10% PEG-4000)", "days in in".

7) Materials & Methods, X-ray crystallography data processing: Missing methods on diffraction data collection.

8) Figure 2, panels B1 and B2: The authors compare crystal and AlphaFold structures of the complex in which the side chain of Arg1076 interacts with Asp10 in the crystal structure while it interacts with a hidden residue in the AlphaFold structure. The author can briefly discuss the two conformations of Arg1076. The authors can also show the hidden residue that interacts with Arg1076 in B2.

9) Figure S1B: The authors should move this panel to the end where they discuss AlphaFold limitations (see my comment #3). In addition, the authors could add a new supplemental figure to show the conservation of the two motifs (see my comment #2).

Reviewer #2 (Remarks to the Author):

The main topic is important: to understand evaluate the performance of AlphaFold2.
The objective is great: to support the assessment with experimental data.
The experimental structures are highly valuable themselves.

However, the manuscript contains statements and interpretations, which are too strong or invalid.

**** Testing individual domains, CEP44 CH and CEP192 Spd2

CEP44:

There are 3 human and 2 yeast CH structures in the PDB dated before AF2 training.
Therefore, the AF2 prediction is not unexpected to match the experimental structure.

“Besides the accurate prediction of the CEP44 N-terminal domain, the AF2 prediction of alternating coiled coil and unstructured regions that follow the CH domain is in good agreement with other prediction methods and provides an easily accessible model of CEP44’s overall domain organisation.”

This should be demonstrated either as a main figure or in the supplementary.

CEP192

Spd2: the domain and associated structures should be described better; 2E6J: NMR structure, thus not included in the training set; 6FVI: released on 2019-03-20, thus were not included in the training set; however, it was included among the templates when generating structures for AF2 database; thus it is not unexpected to match the experimental structure; however: (1) 6FVI is a single domain, so you could put all the emphasis on the novelty of the interface (this interface prediction and the unique insertion are important for AF2 assessment); (2) You could rerun the prediction with disabled template usage that would be a nice assessment of AF2 performance.

“The AF2 model of full-length human CEP192 also provides the first glimpse into its modular organisation.” Similar to other methods, AF2 can not predict the “structure” of disordered/flexible regions. Having a knot is not surprising and does not add to learning about AF2. Because of this issue, AF2 also may not add anything to learning about full-length CEP192. Assessment of other domains than Spd2 may add values.

**** CEP164 and TTBK2 complexes

“While the high-resolution structure of this complex is known (Rosa e Silva et al., 2021)” - not only citation, but PDB id should also be listed.

The a-helix (TTBK approx.1087-1100) and its interactions is not present in the already published complex structure (containing TTBK 1074-1087). So you can not make conclusion on the a-helical region (1087-1100) based on the agreeing 1071-1087 region. It would be highly valuable to have the experimental structure for the complex with longer TTBK segment. If the AF2 prediction is valid than this structure determination should be not more difficult than the previous one with the shorter segment.

In addition, the recent publication of AF2-Multimer suggests that heteromeric complexes should be predicted by the AF2-Multimer and not with AF2.

While tThe analysis assessment of this interaction with TTBK mutations does not exclude the conformation of AF2 predicted complex structure, but these experiments do not support the interface location on CEP164 at all. CEP164 should also be mutated in the interface region to draw a conclusion on the AF2 prediction. (Determining the complex structure could be easier.)
Therefore, I do not think that this is a valid statement: “The AF2 model and our biochemical data provide valuable mechanistic insights into a putative mode of regulation and inspire further

experiments to address these questions.”

You probably want to discuss more AF2 protein-protein and protein-peptide interaction prediction in the introduction or in the discussion. Also, AF2-Multimer.

**** Chibby1-FAM92A complex:

Please change the structure colors of salmon/pink and green to something else for colorblind readers.

“However, our mutational analysis suggests that the binding contribution of this region is smaller than the truncation experiments suggested. Thus, 14-3-3 binding is unlikely to efficiently regulate Chibby1-FAM92A complex formation.”

I think that you should not conclude anything about regulation efficiency through a region based on the the contribution of that region to PPI strength. In spite the smaller contribution of Chibby 1-22 to PPI interaction, it may have an important role in regulation. You simply do not have data on 14-3-3 regulation of the 1-22 Chibby mutant. This is simply too much speculation, should not be formulated, in spite of the following sentence (“Further experiments will be necessary...”).

**** Limitations of using AF2 protein structure predictions

“The general ability of AF2 to correctly and accurately predict protein-protein interactions remains to be determined.”

First, the general ability of AF2 to correctly predict PPIs can not be determined by such a small scale study. Second, AF2-Multimer preprint was published during the review process, so it should be discussed and/or used in this manuscript, too.

“One possible reason for the AF2 under-performance may be due to under-representation of experimentally determined coiled coil structures that are available in the PDB.”

How many coiled-coiled structures are present in the PDB? How much is needed for AF2 for make a good prediction?

“the need to gather more experimental data on these assemblies” -- strongly agreeing

“Another limitation of AF2 is that the predictions yield a static picture” - agreeing, but not novel and surprising (all static methods, such as X-ray or homology modeling suffer from this)

“This prediction is particularly problematic for multidomain proteins in which individual domains are segregated by unstructured regions.” - agreeing, but not novel

This section could sound better if you change its title to Discussion.

The manuscript reads well and only minor errors were observed.

E.g.

from a uncharacterised protein from *P.gingivalis* (pdb 2QSV, probability: 99.53%, e-value: 1.6e-11, identity: 14%), respectively.

... an uncharact... P.<space> gingivalis

Dear Dr. Chong and Dr. Sengupta

First, we would like to thank both reviewers for their constructive feedback that helped us strengthen our manuscript further. We have addressed the points made by the reviewers and modified the manuscript accordingly.

Please find below our point-by-point reply to the reviewers' comments.

Reviewer #1 (Remarks to the Author):

The manuscript entitled "AlphaFold2 and centrosome research: a tale of strengths and limitations" by Breugel, et al describes the application of using AlphaFold2-predicted protein structures for biochemical studies of centrosome proteins. Using X-ray crystallography, the authors validated the accuracy of AlphaFold-predicted structures of CEP44 CH domain and CEP192 domains 4 and 5. They extended their recent structural work on the complex between CEP164 NTD and TTBK2 proline-rich domain, from which they found previously unknown interactions using AlphaFold. They validated such interactions through site-directed mutagenesis. To show whether AlphaFold can generate hypothesis for studying protein/protein interactions, they predicted a complex structure between FAM92A BAR-like domain and Chibby1 coiled-coil domain and validated their interactions using a pull-down assay. Through three examples of six centrosomal proteins, they provide convincing cases that AlphaFold may be broadly applicable for centrosome research. The manuscript is well written, although there are quite a few typos through the text. The work is technically sound and may be of interest to structural biologists, biochemists, and cell biologists in designing and optimizing their experiments. I have some suggestions that need to be addressed prior to acceptance for publication.

1) Results & Discussion, p 4, lines 2-5: The authors state "Besides the accurate prediction, ... with other prediction methods and provides an easily accessible model of CEP44's overall domain organisation." What are the other prediction methods and how were these models compared? The authors should at least give some numbers to justify such a statement.

We have now provided the coiled coil and secondary structure prediction from different algorithms in comparison to the AF2 prediction and show these data in Supplementary Figure 2c in the revised manuscript.

2) Results & Discussion, p 4, second paragraph: " Its most significant sequence conservation is found in two motifs, as judged by multiple sequence alignment of orthologous sequences: Motif 1 (PLXGYGG) located at the C-terminal end of domain 4, and motif 2 (GDEXXR) located within domain 5." Missing either a reference or a multiple sequence alignment to show the conservation of the two motifs. Alternatively, the authors can map the conservation to a surface or a ribbon representation, similar to what they did for CEP44 CH domain (Figure S1A) or CEP 164/TTBK2 (Figure S2D).

We agree that these data would be very useful for the reader. We have now provided a multiple sequence alignment of the Spd2 domain from diverse homologs along with the structure, colour-coded by ConSurf conservation score and with both motifs clearly marked. This is now shown in Supplementary Figure 3a-b in the revised manuscript.

3) Results & Discussion, p 5, lines 2-9: “While the structure of each individual domain is likely in contrast to the knot in the predicted human CEP192 full-length model that is probably a result of erroneous prediction”. Please consider moving this discussion to the section of “Limitations of using AF2 protein structure predictions”.

We have changed to the manuscript and incorporated the corresponding text in the section ‘Limitations of using AF2’ (revised manuscript: Page 9, last paragraph).

4) Results & Discussion, p 7, second line from bottom: Please define “14-3-3” when first time mentioned.

We have now specified which 14-3-3 protein was used (14-3-3ζ) in the corresponding experiment and also provided the gene name (YWHAZ). These changes are found in the revised manuscript on page 8, first paragraph.

5) Conclusion & Outlook: In this reviewer’s option, the authors should focus on Conclusion only. Their Outlook statements are not related to the work they present, and perhaps should be deleted.

As suggested, we have changed the name of this section to Conclusion (revised manuscript: Page 10). We understand that our previous outlook statement might be rather broad. We have made some changes but kept the embedding of our findings in the wider context as we believe it might have some appeal to readers. Thus, we would like to leave this aspect intact, unless the reviewer feels strong about this.

6) Materials & Methods, Protein crystallization: Please correct typos: “10% PEG-4000”, “days in in”.

We have corrected these typos that had escaped our proof reading (revised manuscript: Page 12, third/fourth paragraph). Thanks.

7) Materials & Methods, X-ray crystallography data processing: Missing methods on diffraction data collection.

We have now added this information to the materials and methods section (revised manuscript: Page 12, last paragraph). Thanks for spotting this.

8) Figure 2, panels B1 and B2: The authors compare crystal and AlphaFold structures of the complex in which the side chain of Arg1076 interacts with Asp10 in the crystal structure while it interacts with a hidden residue in the AlphaFold structure. The author can briefly discuss the two conformations of Arg1076. The authors can also show the hidden residue that interacts with Arg1076 in B2.

We have added the information about the corresponding side-chain conformations and also related it to the AlphaFold-multimer prediction. This information is found in the revised manuscript on page 6, first paragraph.

9) Figure S1B: The authors should move this panel to the end where they discuss AlphaFold limitations (see my comment #3). In addition, the authors could add a new supplemental figure to show the conservation of the two motifs (see my comment #2).

We have removed the former panel S1B on the protein knots and only cover this aspect now in the 'Limitations of using AF2' section (revised manuscript: Page 9, last paragraph). See also our reply to the similar point 4 of reviewer 2. As to the conservation of the Spd2 motifs, please see our reply to your point 2 - we provide this information now in Supplementary Figure 3a-b in the revised manuscript.

Reviewer #2 (Remarks to the Author):

The main topic is important: to understand evaluate the performance of AlphaFold2. The objective is great: to support the assessment with experimental data. The experimental structures are highly valuable themselves.

However, the manuscript contains statements and interpretations, which are too strong or invalid.

1) **** Testing individual domains, CEP44 CH and CEP192 Spd2

CEP44:

There are 3 human and 2 yeast CH structures in the PDB dated before AF2 training. Therefore, the AF2 prediction is not unexpected to match the experimental structure.

What is rather unexpected in the CEP44 CH domain AF2 prediction is the outstanding quality of the model that is to atomic detail. We have now pointed this out more clearly in the manuscript (revised manuscript: Page 3, last paragraph). We now also show in the revised manuscript in Supplementary Figure 2a the superpositions of the experimental structure with the AF2 model compared with those of known structures that technically could have been used as templates for homology modelling or molecular replacement to highlight this point further.

2) "Besides the accurate prediction of the CEP44 N-terminal domain, the AF2 prediction of alternating coiled coil and unstructured regions that follow the CH domain is in good agreement with other prediction methods and provides an easily accessible model of CEP44's overall domain organisation."

This should be demonstrated either as a main figure or in the supplementary.

Thanks for pointing this out - please see our reply to Reviewer 1 (point 1). We have now provided the coiled coil and secondary structure prediction from different algorithms compared to the AF2 prediction and show these data in Supplementary Figure 2c in the revised manuscript.

3) CEP192

Spd2: the domain and associated structures should be described better; 2E6J: NMR structure, thus not included in the training set; 6FVI: released on 2019-03-20, thus were not included in the training set; however, it was included among the templates when generating structures for AF2 database; thus it is not unexpected to match the experimental structure; however: (1) 6FVI is a single domain, so you could put all the emphasis on the novelty of the interface (this interface prediction and the unique insertion are important for AF2 assessment); (2) You could rerun the prediction with disabled template usage that would be a nice assessment of AF2 performance.

In our submission manuscript, we had provided the output results from the sequence searches using HHpred (revised manuscript: Page 3, first paragraph ; Page 4, last paragraph). The most similar to the Spd2 domain in sequence is the structure of an uncharacterized protein from *P. gingivalis* (PDB code 2qsv) and its crystal structure has been included in the AF2 training set. This structure also consists of two PapD-like domains and the HHpred alignment expands to both domains but terminates after CEP192 Spd2 domain strand beta-f. Please refer to the new Supplementary Figure 3c in the revised manuscript for more detailed information: The 2qsv structure has a very different orientation of domains compared to the Spd2 domain and does not contain its 60 residues insertion. Below are some data from superpositions and sequence analysis for the reviewer's consideration.

Structure superposition with Topmatch:

2qsv superposed on CEP192 dom4 – 100 structurally equivalent residues, 2.48 rmsd , 11 identical residues

2qsv superposed on CEP192 dom5 – 86 structurally equivalent residues, 1.88 rmsd , 15 identical residues

6fvi superposed on CEP192 dom4 – 94 structurally equivalent residues, 2.53 rmsd , 8 identical residues

6fvi superposed on CEP192 dom5 – 99 structurally equivalent residues, 2.26 rmsd , 10 identical residues

HHpred list of the high scoring database matches:

2qsv - Probability: 99.61%, E-value: 2.7e-12, Score: 104.31, Identities: 13%

6fvi - Probability: 98.67%, E-value: 0.0000034, Score: 65.03, Identities: 15%

2e6j - Probability: 98.58%, E-value: 0.0000042, Score: 60.63, Identities: 19%

2ys4 - Probability: 98.36%, E-value: 0.000059, Score: 55.78, Identities: 19%

We have now provided in Supplementary Figure 3d-e of the revised manuscript the structure superpositions of the two CEP192 Spd2 subdomains (domain 4 and 5) with these structures and also provided the corresponding structure-based sequence alignments. Furthermore, we have pointed out these similarities in the text along the lines suggested by you, i.e. focussing more on the novelty of the interface region (revised manuscript: Page 4, last paragraph).

The rerun of the AF2 prediction for CEP192 domain 1-6 (i.e. including its Spd2 domain, see also our response to your point 4) gave essentially the same result for the Spd2 domain structure, despite the corresponding AF2 Colab not using templates. We point this out in the revised manuscript (page 9, last paragraph and page 13, paragraph 5).

4) “The AF2 model of full-length human CEP192 also provides the first glimpse into its modular organisation.” Similar to other methods, AF2 can not predict the “structure” of disordered/flexible regions. Having a knot is not surprising and does not add to learning about AF2. Because of this issue, AF2 also may not add anything to learning about full-length CEP192. Assessment of other domains than Spd2 may add values.

In the past, knots have been considered impossible and topological constraints have been used to prevent knots in the protein models. As we pointed out in the manuscript, currently there is a growing number of proteins having genuine knots. For a casual reader without insights into this problem we consider that this is informative and important as it warns and draws attention to this problem.

When we reran six consecutive domains of CEP192 (domain 1-6) with the latest AF2, we could not detect any knots, unlike in the full length CEP192 AF2 prediction available from the AF2 Protein Structure Database. Thus, we believe that the problem might originate either from the length of the predicted protein or the exact way the AF2 database has been computed.

We have now removed the corresponding knot figure in the supplement, as this is a more minor point, but kept the corresponding (shortened and updated) discussion of the protein knot occurrences in the section describing the AF2 limitations (page 9, last paragraph in the revised manuscript).

5) **** CEP164 and TTBK2 complexes

“While the high-resolution structure of this complex is known (Rosa e Silva et al., 2021)” - not only citation, but PDB id should also be listed.

Thanks for spotting this - we had put the PDB code in the figure legend, but now have included it also in the main body (page 5, fourth paragraph in the revised manuscript).

6) The α -helix (TTBK approx. 1087-1100) and its interactions is not present in the already published complex structure (containing TTBK 1074-1087). So you can not make conclusion on the α -helical region (1087-1100) based on the agreeing 1071-1087 region. It would be highly valuable to have the experimental structure for the complex with longer TTBK segment. If the AF2 prediction is valid than this structure determination should be not more difficult than the previous one with the shorter segment. In addition, the recent publication of AF2-Multimer suggests that heteromeric complexes should be predicted by the AF2-Multimer and not with AF2.

The AF2-Multimer prediction gave similar results and we discuss the corresponding findings now in the revised manuscript (page 5, last paragraph - page 6, first paragraph).

The experimental structure of the smaller complex was obtained when bound to nanobodies that were required for crystallisation. Due to the CEP164 binding site of these nanobodies that co-localizes with that of the TTBK2 1087-1100 sequence, obtaining similar crystals of the complex with longer TTBK2 fragments is unfortunately not possible. Our attempts to get protein crystals of the larger complex without nanobodies have not been successful so far (see also our reply to your point 7).

7) While the analysis assessment of this interaction with TTBK mutations does not exclude the conformation of AF2 predicted complex structure, but these experiments do not support the interface location on CEP164 at all. CEP164 should also be mutated in the interface region to draw a conclusion on the AF2 prediction. (Determining the complex structure could be easier.) Therefore, I do not think that this is a valid statement: "The AF2 model and our biochemical data provide valuable mechanistic insights into a putative mode of regulation and inspire further experiments to address these questions." You probably want to discuss more AF2 protein-protein and protein-peptide interaction prediction in the introduction or in the discussion. Also, AF2-Multimer.

Similar to the other AF2 predictions in the manuscript, we have now confirmed the models with AF2- Multimer and also discuss this in the revised manuscript (page 5, last paragraph - page 6, first paragraph).

As to mutational analysis of CEP164, this is complicated by the fact that the corresponding part of CEP164 also has a critical role in ensuring WW domain integrity (Rosa e Silva et al., 2021). Thus, as done in the manuscript, targeting the binding region in TTBK2 (that is strongly predicted to be disordered when unbound) therefore is the best option to confirm the AF2 predictions. However, while the AF2 model and our biochemical data (from TTBK2 mutations and truncations) are strongly suggestive of the existence of this additional interface, we agree with the reviewer that it is wise to tone down the corresponding section of the manuscript. We have modified the sections in question to the following, more cautious statements:

"Our biochemical data together with the associated AF2 model provide valuable insights and create hypotheses that can be further explored experimentally to test this proposed regulatory mechanism." (page 6, last paragraph in the revised manuscript).

And:

"However, further research will be necessary to structurally confirm the AF2 prediction and to establish whether PtdIns4P binding can affect the identified CEP164-TTBK2 interface and regulate protein complex formation and ciliogenesis *in vivo*." (page 6, last paragraph in the revised manuscript).

As explained in our reply to your point 6, we cannot use the nanobodies as crystallisation chaperones that we employed to obtain protein crystals of the smaller CEP164-TTBK2 complex. Our attempts to obtain protein crystals without the use of nanobodies have not been successful so far, probably because the crystal packing interactions from the nanobodies are missing.

8) **** Chibby1-FAM92A complex:

Please change the structure colors of salmon/pink and green to something else for colorblind readers.

That is something we have been overlooked - thanks for pointing this out. We have now changed the colours of our panels to colour combinations that are more friendly to colour-blind readers (Figure 1b,c, Figure 2b, Figure 3a, Supplementary Figure 2b in the revised manuscript).

9) “However, our mutational analysis suggests that the binding contribution of this region is smaller than the truncation experiments suggested. Thus, 14-3-3 binding is unlikely to efficiently regulate Chibby1-FAM92A complex formation.”

I think that you should not conclude anything about regulation efficiency through a region based on the the contribution of that region to PPI strength. In spite the smaller contribution of Chibby 1-22 to PPI interaction, it may have an important role in regulation. You simply do not have data on 14-3-3 regulation of the 1-22 Chibby mutant. This is simply too much speculation, should not be formulated, in spite of the following sentence (“Further experiments will be necessary...”).

We show that the N-terminal region of Chibby1 is of a more minor importance for the Chibby1-Fam92A interaction. We believe that this makes the corresponding interface not the most obvious one for regulating this complex. However, we agree with the reviewer that we cannot exclude this possibility, especially in the absence of further experimental *in vivo* data. Thus, as suggested, we changed the corresponding passage to:

“However, our mutational analysis suggests that the binding contribution of this region is smaller than the truncation experiments indicated. Further experiments will be required to evaluate whether and how Chibby1 phosphorylation and 14-3-3 binding can impact on FAM92A engagement with Chibby1 and cilia formation *in vivo*” (page 8, first paragraph in the revised manuscript).

10) **** Limitations of using AF2 protein structure predictions

“The general ability of AF2 to correctly and accurately predict protein-protein interactions remains to be determined.” First, the general ability of AF2 to correctly predict PPIs can not be determined by such a small scale study. Second, AF2-Multimer preprint was published during the review process, so it should be discussed and/or used in this manuscript, too.

We have not claimed in the manuscript that our study provides a test of AF2’s general ability to predict PPIs. We portray our study as showcasing AF2’s powerful capabilities, with a focus on the centrosome / centriole field and we believe that our findings as such will be of interest for this field but also beyond.

As to AF2-Multimer, as pointed out previously, we have confirmed the structure predictions with it and commented on this in the revised manuscript (page 6, first paragraph; page 7, second paragraph; page 13, third and fourth paragraph).

11) “One possible reason for the AF2 under-performance may be due to under-

representation of experimentally determined coiled coil structures that are available in the PDB.”

How many coiled-coiled structures are present in the PDB? How much is needed for AF2 for make a good prediction?

According to the latest version of SCOP 2 (<https://scop.mrc-lmb.cam.ac.uk/>), there are 470 domains in the fibrous proteins category (mainly containing coiled coil domains) compared to 34094 globular and 1119 membrane domains. SCOP 2 may not have a 100% coverage of the PDB but the numbers clearly show the trend. We point this now out in the revised manuscript (page 8, last paragraph - page 9, first paragraph).

12) “the need to gather more experimental data on these assemblies” — strongly agreeing

“Another limitation of AF2 is that the predictions yield a static picture” - agreeing, but not novel and surprising (all static methods, such as X-ray or homology modeling suffer from this)

“This prediction is particularly problematic for multidomain proteins in which individual domains are segregated by unstructured regions.” - agreeing, but not novel

This section could sound better if you change its title to Discussion.

We agree with the reviewer, but have not stated in our manuscript that these are particularly novel aspects. To make that clearer we toned down the passage on inter-domain contact prediction by adding a corresponding qualifier and also by linking the protein dynamics aspect further to our findings (page 9, third paragraph and second paragraph, respectively). As pointed out in our reply to Reviewer 1 (point 5) we have changed the name of the last section to “Conclusion” (revised manuscript: Page 10).

13) The manuscript reads well and only minor errors were observed. E.g. from a uncharacterised protein from *P.gingivalis* (pdb 2QSV, probability: 99.53%, e-value: 1.6e-11, identity: 14%), respectively.

... an uncharact... P.<space> gingivalis

Thanks for spotting this - we have corrected this typo in the revised manuscript (page 3, first paragraph).

References

Rosa E Silva I, Binó L, Johnson CM, Rutherford TJ, Neuhaus D, Andreeva A, Čajánek L, van Breugel M. (2021). Molecular mechanisms underlying the role of the centriolar CEP164-TTBK2 complex in ciliopathies. *Structure* S0969-2126(21)00302-6.

REVIEWERS' COMMENTS:

Reviewer #2 (Remarks to the Author):

In the revised manuscript, the authors have addressed most of my concerns. The only comment I have is that the authors mixed discussion in their last section of "Conclusion".

P.10, lines 350-364. Without supporting data, the authors speculate on future applications of AF2 in structural biology and life sciences.

The authors should consider one of the three suggestions:

- 1) Delete the "Conclusion" section.
- 2) Find a more appropriate title for this section.
- 3) Remove text (lines 350-364) that does not belong to "Conclusion".

Reviewer #3 (Remarks to the Author):

Thanks for the detailed answers and explanations.

Dear Dr. Chong and Dr. Sengupta

We would like to thank both reviewers for going through our revised manuscript. We now addressed the last point raised by reviewer #2 and modified the manuscript accordingly.

Please find below our point-by-point reply to the reviewers' comments.

REVIEWERS' COMMENTS:

Reviewer #2 (Remarks to the Author):

In the revised manuscript, the authors have addressed most of my concerns. The only comment I have is that the authors mixed discussion in their last section of "Conclusion".

P.10, lines 350-364. Without supporting data, the authors speculate on future applications of AF2 in structural biology and life sciences.

The authors should consider one of the three suggestions:

- 1) Delete the "Conclusion" section.
- 2) Find a more appropriate title for this section.
- 3) Remove text (lines 350-364) that does not belong to "Conclusion".

We followed your suggestion 3) and removed the corresponding text from the Conclusion section in the revised manuscript (Page 10).

Reviewer #3 (Remarks to the Author):

Thanks for the detailed answers and explanations.